# Phlpp1 is induced by estrogen in osteoclasts and its loss in Ctsk-expressing cells does not protect against ovariectomy-induced bone loss

**Marcelline K. Hanson**[1☯], **Ismael Y. Karkache**[1☯], **David H. H. Molstad**[1], **Andrew A. Norton**[3], **Kim C. Mansky**[3], **Elizabeth W. Bradley**[1,2]*

**1** Department of Orthopedic Surgery, University of Minnesota, Minneapolis, MN, United States of America, **2** Stem Cell Institute, University of Minnesota, Minneapolis, MN, United States of America, **3** Department of Developmental and Surgical Sciences, University of Minnesota, Minneapolis, MN, United States of America

☯ These authors contributed equally to this work.

* ebradle1@umn.edu

## Abstract

Prior studies demonstrated that deletion of the protein phosphatase Phlpp1 in Ctsk-Cre expressing cells enhances bone mass, characterized by diminished osteoclast activity and increased coupling to bone formation. Due to non-specific expression of Ctsk-Cre, the definitive mechanism for this observation was unclear. To further define the role of bone resorbing osteoclasts, we performed ovariectomy (Ovx) and Sham surgeries on Phlpp1 cKO$_{Ctsk}$ and WT mice. Micro-CT analyses confirmed enhanced bone mass of Phlpp1 cKO$_{Ctsk}$ Sham females. In contrast, Ovx induced bone loss in both groups, with no difference between Phlpp1 cKO$_{Ctsk}$ and WT mice. Histomorphometry demonstrated that Ovx mice lacked differences in osteoclasts per bone surface, suggesting that estradiol (E2) is required for Phlpp1 deficiency to have an effect. We performed high throughput unbiased transcriptional profiling of Phlpp1 cKO$_{Ctsk}$ osteoclasts and identified 290 differentially expressed genes. By cross-referencing these differentially expressed genes with all estrogen response element (ERE) containing genes, we identified IGFBP4 as potential estrogen-dependent target of Phlpp1. E2 induced PHLPP1 expression, but reduced IGFBP4 levels. Moreover, genetic deletion or chemical inhibition of Phlpp1 was correlated with IGFBP4 levels. We then assessed IGFBP4 expression by osteoclasts in vivo within intact 12-week-old females. Modest IGFBP4 immunohistochemical staining of TRAP$^+$ osteoclasts within WT females was observed. In contrast, TRAP$^+$ bone lining cells within intact Phlpp1 cKO$_{Ctsk}$ females robustly expressed IGFBP4, but levels were diminished within TRAP$^+$ bone lining cells following Ovx. These results demonstrate that effects of Phlpp1 conditional deficiency are lost following Ovx, potentially due to estrogen-dependent regulation of IGFBP4.

## Introduction

Bone loss is a natural phenomenon of the aging process. As both men and women age, there is a relative imbalance in bone resorption as compared to formation which leads to negative

**Data Availability Statement:** The RNA-Seq data has been deposited in the Array Express public database (accession# E-MTAB-10487).

**Funding:** EWB received a research grant from the National Institutes of Health, National Institute for Arthritis and Skin Diseases (AR072634, www. niams.nih.gov). The funders had no role in study design, data collection and analysis, decision to publish, or preparation of the manuscript.

**Competing interests:** The authors have declared that no competing interests exist.

bone balance. Osteoporosis is a bone disorder marked by low bone mass and low bone quality. This leads to compromised bone strength, hence increased ease of fracture. Osteoporotic fractures are prevalent around the world and are a major cause of morbidity and mortality in the elderly [1]. The underlying causes of osteoporosis are subjective and multifactorial, influenced by diet, physical activities, hormonal status, cytokines, and overall health status such as diabetes mellitus and glucocorticoid treatment. Regardless of the etiology, osteoporosis is characterized by both structural and cellular changes to cancellous bone and endocortical surface. For women, bone loss is accentuated during the perimenopausal and postmenopausal periods, punctuated by rapid decline in endogenous estrogen. Estrogen deficiency is strongly correlated with the acceleration of cancellous bone loss and the decrease of cortical bone in women [2]. Moreover, estrogen deficiency enhances bone resorption. In contrast, hormone replacement therapy in pre-clinical ovariectomy models or menopausal women has been shown to reduce bone loss and risk of fractures in the vertebrae as well as in non-vertebral sites including the hip, adding to evidence that estrogen is critical to maintaining bone mass [3].

The process of bone remodeling is carried out by the coupled activity of bone resorbing osteoclasts and bone forming osteoblasts. There are three phases of the bone remodeling process: resorption, reversal and formation. In a balanced system, the amount of bone removed by osteoclasts exactly matches the amount laid down by the osteoblast. However, age-related decline disrupts the reversal phase, leading to uncoupling of bone resorption and bone formation phases. This creates an imbalance between the amount of bone resorbed compared to that formed with each remodeling cycle. The compounded effects of this whole process are exacerbated by an increase in the frequency with which new remodeling cycles are activated after the menopause. This leads to postmenopausal acceleration of bone loss [4].

Phlpp1 (PH domain and leucine rich protein phosphatase 1) belongs to a class of metal-dependent phosphatases [5]. Phlpp1 limits the activity of anabolic kinases including Akt2, Raf, and typical and atypical PKC isoforms [5]. We previously demonstrated that germline deletion of protein phosphatase Phlpp1 limits bone mass [6], but we could not discern the cell type specific functions within this model. Conversely, conditional deletion of Phlpp1 in Cathepsin K (Ctsk)-Cre-expressing cells resulted in enhancement of bone mass characterized by diminished osteoclast activity and enhanced coupling to bone formation [7]. Due to the limitations of the Ctsk-Cre driver, including expression within mesenchymal lineage cells [8–12], we further explored the functions of Phlpp1 in a model of enhanced bone resorption. In this study, we determined if Phlpp1 cKO$_{Ctsk}$ mice were protected from bone loss as a result of ovariectomy in this study. We find that Phlpp1 cKO$_{Ctsk}$ mice are not protected from ovariectomy induced bone loss. Furthermore, this study demonstrates that Phlpp1 limits expression of Ifgbp4, known to be required for optimal bone mass attainment in females [13], and that this repression is lost following ovariectomy.

## Methods

### Generation of Phlpp1 conditional knockout mice

We previously described generation of mice harboring the Phlpp1 floxed allele [7] used in this study. Ctsk-Cre driver mice used in this study were obtained from R. A. Davey and were previously described [8] Phlpp1$^{fl/+}$ mice were mated with mice expressing Cre-recombinase under the control of the Ctsk promoter [7, 8]. Mice were genotyped for Cre [14] or the Phlpp1-floxed allele using the following primers: forward: 5′–CAGTGGATATCTGGATAATC–3′, reverse: 5′–GATGAGTGTTTTCATGAGGA–3′. Conditional knockout animals from these crossings are denoted as Phlpp1 cKO$_{Ctsk}$ mice and are on the C57Bl/6 background. Cre$^+$ control littermates from crossings were used as controls as appropriate. Animals were housed in an accredited

facility under a 12-h light/dark cycle and provided water and food *ad libitum*. Ovariectomy surgeries were performed on 12-week-old Phlpp1 cKO$_{Ctsk}$ females (n = 4) or their control Cre$^+$ littermates (n = 5) as previously described [15]. Sham surgeries were also performed on 12-week-old Phlpp1 cKO$_{Ctsk}$ females (n = 10) or Cre$^+$ littermates (n = 9) for a total of n = 29 mice. Mice were anesthetized using 1–2% isoflurane delivered via inhalation prior to surgery. Warm saline was administered subcutaneously immediately after surgery. Animals were observed for one hour following surgery before being returned to housing. Following return to housing, animals were observed once per day for three days and then once per week for the duration of 4 weeks post-surgery. The incision site was monitored for signs of infection. Mice were also be monitored for pain following surgeries using body condition scoring. All animal research was conducted according to guidelines provided by the National Institute of Health and the Institute of Laboratory Animal Resources, National Research Council. The University of Minnesota Institutional Animal Care and Use Committee approved all animal studies Mice were euthanized by CO2 asphyxiation, using cervical dislocation as a secondary measure of euthanasia.

## Micro-computed tomography

Femora from ovariectomized mice were collected 4 weeks post-ovariectomy surgery and fixed in 10% neutral buffered formalin for 48 h, then stored in 70% ethanol. Blinded study staff performed scanning at 70 kV, 221 ms with a 10.5-μm voxel size using a Scanco Viva40 micro-CT. For cortical bone analyses, a region of interest was defined at 10% of total femur length beginning at the femoral midpoint; defining the outer cortical shell and running a midshaft analysis with 260-threshold air filling correction analyzed samples. For trabecular measurements, a region of interest was defined at 10% of total femur length starting immediately proximal to the growth plate; samples were analyzed using a 220-threshold air filling correction.

## Histomorphometry and immunohistochemistry

Tibiae were collected from ovariectomized mice 4 weeks post-surgery and fixed formalin for 48 h, then stored in 70% ethanol. Tibiae were then decalcified in 15% EDTA for 14 days. Tissues were paraffin embedded and 7-micron sections were generated and TRAP/fast green stained [7]. Standardized histomorphometry was performed as previously described [7]. IHC staining was performed with antibodies directed to Insulin-like growth factor binding protein 4 (Igfbp4) (Millipore, #06–109) or with an isotype control IgG. Detection was accomplished using the Mouse and Rabbit Specific HRP (ABC) Detection IHC Kit (Abcam, #ab64264) using the substrate 3,3′-diaminobenzidine (Sigma Aldrich, St. Louis, MO). Sections were co-stained with TRAP/Fast Green stain as previously described [7].

## Osteoclast differentiation

Hind limbs were dissected and bone marrow macrophages were collected from female 6 to 8-week-old Control, Phlpp1 cKO$_{Ctsk}$ or Phlpp1$^{-/-}$ mice as previously described [16]. Briefly, cells were flushed, pelleted and red blood cells were lysed (RBC Lysis Buffer, #00-4333-57, Invitrogen, Carlsbad, CA). Cells were pelleted and cultured overnight in phenol red-free alpha MEM, 10% FBS and 1% antibiotic/antimycotic supplemented with 35 ng/mL M-CSF (#416-ML, R&D, Minneapolis, MN). Non-adherent cells were then placed in culture medium supplemented with 35 ng/mL M-CSF, and 90 ng/mL RANKL (#315–11, Prepro Tech, Rocky Hill, NJ). Cultures were fed on day 3 with culture medium plus 35 ng/ml M-CSF, and 90 ng/mL RANKL and treated as described within the text and figures. Each value shown is determined in triplicate and repeated three times. Shown is the average.

## TRAP staining

TRAP staining was performed as previously described [7]. Briefly, cells were fixed on cover glass with 10% neutral buffered formalin for 10 minutes and then washed 3 times with phosphate-buffered saline (PBS) [7]. Fixed cells were TRAP stained using the Acid Phosphatase, Leukocyte (TRAP) Kit (#387A-1KT, Sigma-Aldrich) and mounted to slides using Vectashield with DAPI (#H-1200, Vector Laboratories, Burlingame, CA) [7]. For osteoclastogenesis experiments, three cover glasses were used per experimental condition. For each cover glass, three fields were imaged using a 10X objective [7]. Images were digitally photographed. Osteoclasts were defined as TRAP$^+$ cells with 3 or more nuclei [7]. Each experiment was repeated independently, each three times. Shown is the average.

## RNA-sequencing and bioinformatics analyses

Bone marrow macrophages were collected from 6-week-old female Phlpp1 cKO$_{Ctsk}$ mice or their wild type littermates and placed in osteoclastogenic conditions as previously described [7]. Primary osteoclasts were then lysed in TRIzol (Invitrogen) and total RNA was collected. High-throughput RNA-Sequencing was performed using RNA from day 4 Phlpp1 cKO$_{Ctsk}$ or wild type osteoclasts as previously reported [17–19]. Briefly, after read alignment, paired-end reads are aligned by TopHat 2.0.6 against the mm10 genome using the bowtie1 aligner option [19, 20]. RPKM values for gene lists were filtered for anything $\geq$0.1 and a fold change $\geq$2, excluding microRNAs and small nuclear RNAs. Data are deposited in the Array Express public database (accession #489556). The resulting gene lists (290 differentially expressed genes) were cross-referenced with all ERE-containing genes within the within the database defined by Bourdeau et al. [21] to obtain a list of 67 potential estrogen-dependent targets of Phlpp1.

## Western blotting

Cells were placed on ice, rinsed twice with PBS and lysed in a buffered SDS solution (0.1% glycerol, 0.01% SDS, 0.1 M Tris, pH 6.8). The BioRad Dc assay was performed, and 40 micrograms of total protein from each sample were resolved by SDS-PAGE. Proteins were transferred to polyvinylidene difluoride membrane which was subsequently blocked with 5% non-fat milk in TBS+0.1% Tween. Western blotting was performed with antibodies (1:1000) for Phlpp1 (Millipore, #07–1341), Igfbp4 (Millipore, #06–109), phospho-Ser473 Akt (Cell Signaling Technology, #4332), Akt (Cell Signaling Technology, #2920), Histone 3 (Abcam, #ab176840) or Tubulin (Developmental Studies Hybridoma Bank, E7) and corresponding secondary antibodies conjugated to horseradish peroxidase (Cell Signaling Technology, #7074 and #7076). Antibody binding was detected with the Supersignal West Femto Chemiluminescent Substrate (#34096, Pierce Biotechnology, Rockford, IL). Resulting bands were collected via radiography and digitally scanned. Each experiment was repeated at least three times reflecting the average of these experiments.

## Statistics

Data shown are the mean ± standard deviation. A Student's t test was performed when one experimental comparison was made. For experiments requiring multiple comparisons, a one-way analysis of variance was performed. Differences with a $p < 0.05$ were considered statistically significant. All analyses were performed using GraphPad Prism 8 software.

## Results

### Phlpp1 deletion within Ctsk-expressing cells does not protect against ovariectomy-induced bone loss

Conditional deletion of Phlpp1 in Ctsk-Cre expressing cells increases osteoclastogenesis, but enhances bone mass [7]. Elevated bone mass within this model was attributed to decreased osteoclast activity and increased coupling to bone formation. Because Ctsk-Cre is expressed by both bone resorbing osteoclasts, as well as mesenchymal lineage cells [8–12], the effects of Phlpp1 conditional deficiency could be attributed to either or both cell lineages. To understand the role of Phlpp1 in a model of enhanced bone resorption, we assessed the effects of Phlpp1 conditional deletion using the Ctsk-Cre driver in an ovariectomy model. Ovariectomy or Sham surgeries were performed on 12-week-old Phlpp1 cKO$_{Ctsk}$ females or the sex-matched control Cre+ littermates. Micro-CT analyses demonstrated that Sham operated Phlpp1 cKO$_{Ctsk}$ females exhibited increased bone parameters, including BV/TV, trabecular number, trabecular thickness and diminished trabecular spacing four weeks post-surgery (Fig 1). While both control and Phlpp1 cKO$_{Ctsk}$ mice lost bone in response to ovariectomy, no significant difference was detected between these two groups (Fig 1). Bone histomorphometric analyses confirmed that control and Phlpp1 cKO$_{Ctsk}$ mice subjected to ovariectomy surgeries do not have statistically significant differences in BV/TV or osteoclasts per bone surface (Fig 2A). Moreover, the percent change in bone mass when comparing Sham to ovariectomized mice of each genotype was unaltered (Table 1). In contrast, Phlpp1 cKO$_{Ctsk}$ females demonstrated a greater

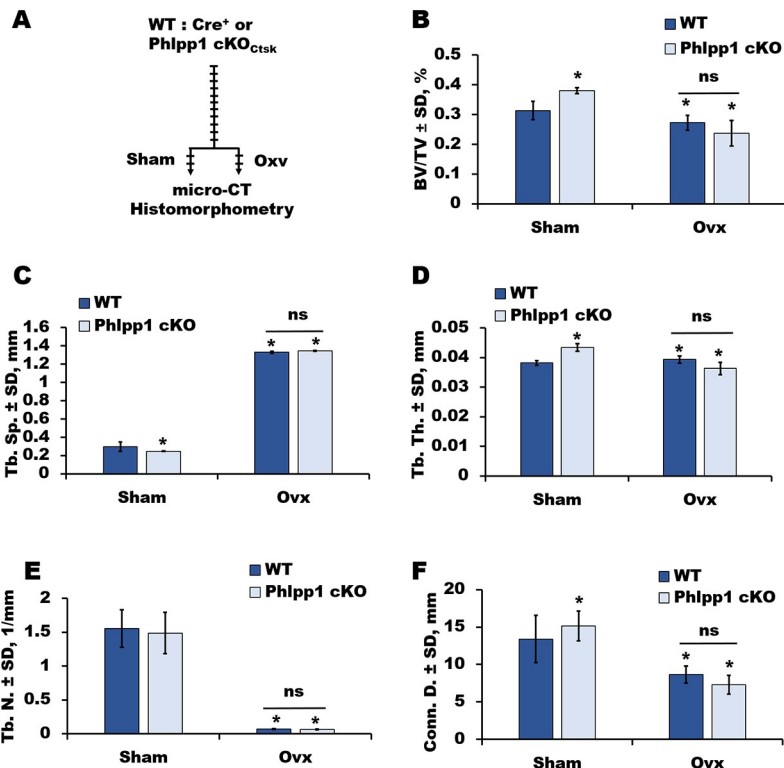

**Fig 1. Phlpp1 cKO$_{Ctsk}$ mice are not protected from ovariectomy-induced bone loss.** (A-F) Ovariectomy or Sham surgeries were performed on Phlpp1 cKO$_{Ctsk}$ 12-week-old females and their wild-type littermates. Four weeks after surgery, the right femora were collected for analyses. (A) Experimental overview. Femora were collected, scanned via micro-CT and evaluated for (B) bone volume / total volume (BV/TV), (C) trabecular spacing (Tb. Sp.), (D) trabecular thickness (Tb. Th.), (E) trabecular number (Tb. N.), and (F) connective density (Conn D). *p<0.05.

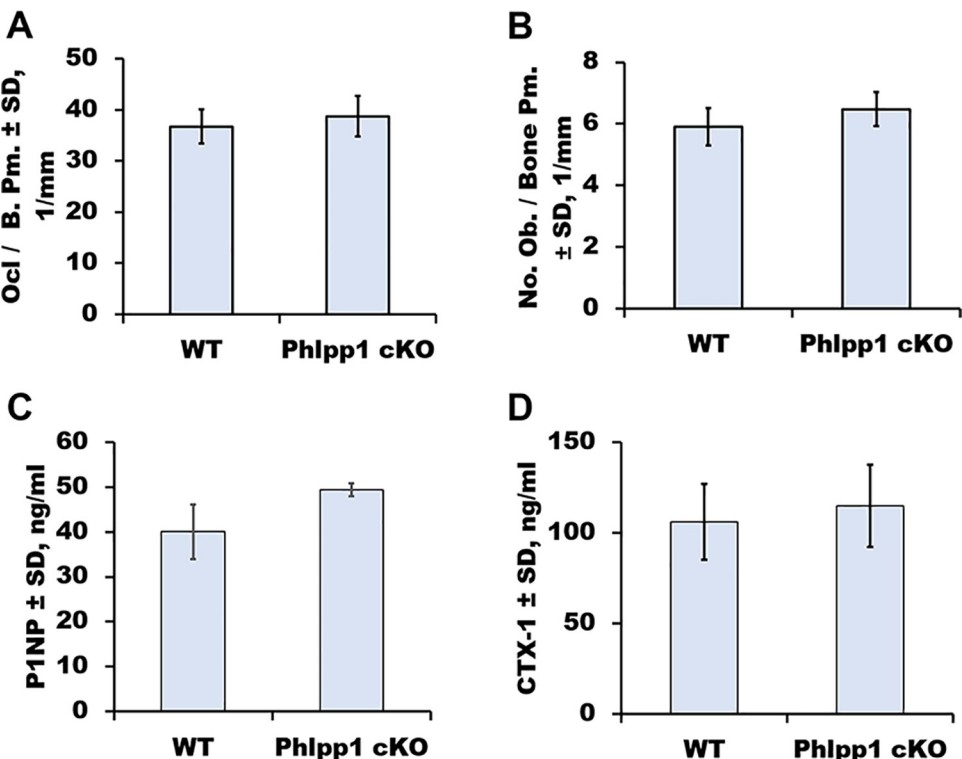

**Fig 2. Phlpp1 cKO_Ctsk mice are not protected from ovariectomy-induced changes in osteoclast number.** (A, B) Ovariectomy surgeries were performed on Phlpp1 cKO_Ctsk 12-week-old females and their wild-type littermates. Four weeks after surgery, the right tibiae were collected for histomorphometric analyses. (A) Osteoclasts per bone perimeter (Ocl / B. Pm.), (B) osteoblasts per bone perimeter (Ob. Pm. / B. Pm.). Serum ELISAs for P1NP (C) and CTX-1 (D) were performed.

percent change in trabecular thickness and spacing (Table 1). Osteoblast number per bone perimeter was also unchanged (Fig 2B). No changes in serum markers of bone resorption (CTX, Fig 2C) or bone formation (P1NP, Fig 2D) were noted. These results demonstrate that Phlpp1 cKO_Ctsk mice do not retain enhanced bone production following ovariectomy.

### Igfbp4 is an ERE containing gene that is upregulated by Phlpp1 deficient cells

Since Phlpp1 conditional deficiency did not protect against ovariectomy-induced bone loss, this suggests that intact estrogen status is required for the effects of Phlpp1 deficiency on bone mass. We therefore, sought to identify the factor(s) that Phlpp1 may regulate in an estrogen-dependent fashion. Osteoclasts were differentiated ex vivo using bone marrow macrophages derived from 4-6-week-old Phlpp1 cKO_Ctsk females or their control Cre+ littermates. Unbiased, high throughput RNA sequencing was performed using total RNA derived from these

**Table 1. Percent change in bone parameters between control and Phlpp1 cKO_Ctsk mice following ovariectomy.**

| | % Change, BV/TV | % Change, Tb. Th. | % Change, Tb. Sp. | % Change, Tb. N. | % Change, Conn. D. |
|---|---|---|---|---|---|
| **WT** | 86.78 ± 8.13 | 102.95 ± 3.13 | 449.96 ± 4.26 | 117.00 ± 0.47 | 4.47 ± 8.36 |
| **Phlpp1 cKO_Ctsk** | 62.28 ± 11.27 | 83.61 ± 4.82 | 545.42 ± 1.68 | 110.63 ± 0.53 | 4.29 ± 8.27 |
| **p Value** | 0.11 | 0.01* | $3 \times 10^{-7}$* | 0.80 | 0.21 |

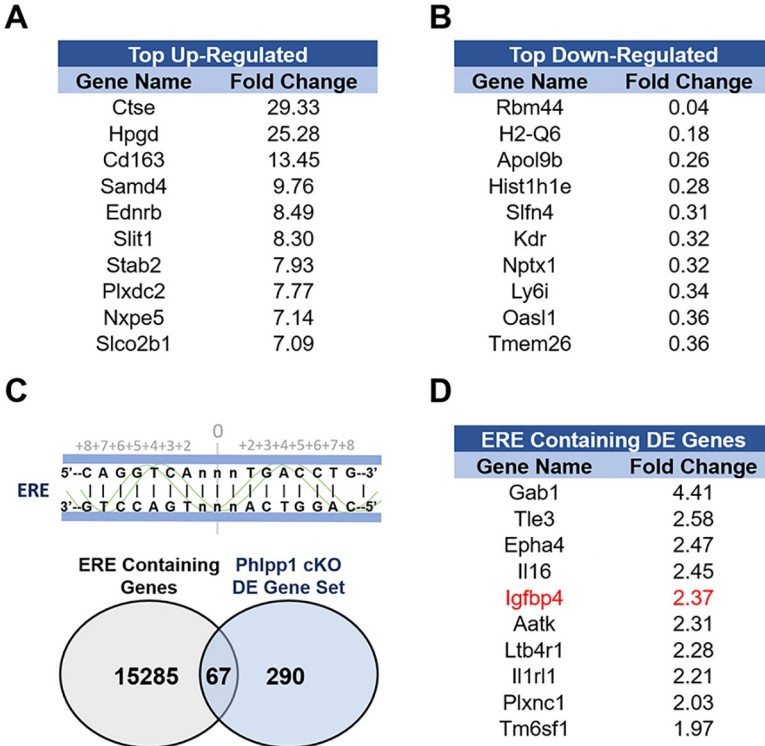

**Fig 3. Identification of ERE-containing genes that are differentially expressed by Phlpp1 cKO_Ctsk osteoclasts.**
RNA from female day 4 Phlpp1 cKO_Ctsk osteoclasts or wild type counterparts was collected and used for high-throughput RNA sequencing. Listed genes were either (A) upregulated or (B) downregulated by two-fold or greater with an FDR < 0.05. (C) Venn diagram illustrating Phlpp1 cKO_Ctsk differentially regulated genes that also contain an ERE. (D) List of the top Phlpp1 cKO_Ctsk differentially expressed ERE-containing genes.

cultures and the top up-regulated (Fig 3A) and down-regulated (Fig 3B) genes were identified. These 290 differentially expressed genes were cross-referenced with an ERE-containing gene database [21] and 67 overlapping genes were identified (Fig 3C). The top differentially expressed ERE-containing genes are listed in Fig 3D. One of the top ERE-containing differentially regulated genes was *Igfbp4*, a gene previously identified to control bone mass in a sex-specific manner [13].

## Phlpp1 deficiency alters Igfbp4 in an estrogen-dependent fashion

Igfbp4 germline deficiency reduces bone mass in females; thus, we hypothesized that Phlpp1 represses expression of Igfbp4 in an estrogen-dependent fashion. We collected osteoclast progenitor cells from 4-6-week-old female Phlpp1 cKO_Ctsk mice or their sex-matched littermates. Reduced PHLPP1 levels were correlated with enhanced Akt phosphorylation and levels of IGFBP4 (Fig 4A). We next assessed the effects of PHLPP1 inhibition using the small molecule inhibitor NSC 117079. Osteoclast progenitor cells were collected from wild type 4-6-week-old female mice and exposed to 5 μM NSC 117079 or vehicle for 24 hours. Enhanced Akt phosphorylation was accompanied by elevated IGFBP4 levels (Fig 4B). We also determined how estradiol affected PHLPP1 and IGFBP4 expression. Osteoclast progenitor cells were collected from wild type or Phlpp1$^{-/-}$ 4-6-week-old female mice and treated with 2 nM estradiol as shown in Fig 4C. While estradiol induced PHLPP1 levels within the time course, IGFBP4 levels diminished (Fig 4C). Although IGFBP4 levels were elevated, estradiol did not reduce IGFBP4

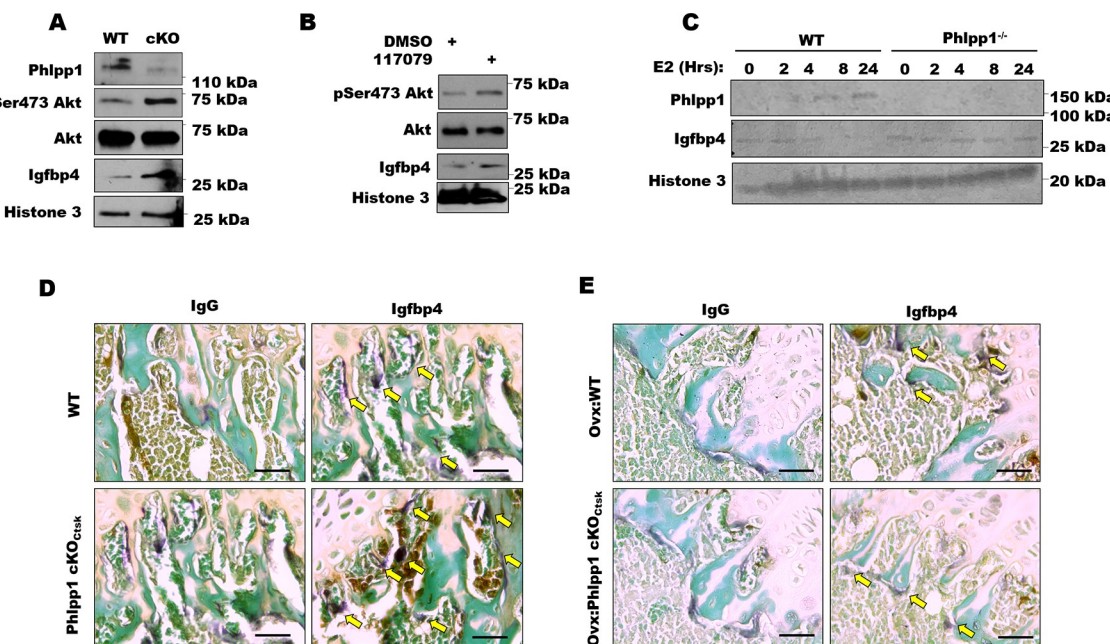

**Fig 4. Estrogen induces Phlpp1 expression, but represses Igfbp4 expression.** (A) Osteoclasts progenitors were collected from 4-6-week-old Phlpp1 cKO$_{Ctsk}$ females or their littermate controls. Western blotting was performed. (B) Osteoclast progenitors were collected from C57Bl/6 mice and cultured in the presence of the Phlpp inhibitor NSC 117079 (5μM) or vehicle control for 24 hours. Western blotting was performed. (C) Osteoclast progenitors were collected from control or Phlpp1$^{-/-}$ female mice were treated with 2 ng/ml estradiol (E2) for the indicated times and western blotting was performed. (D) Tibiae were collected from 12-week-old Phlpp1 cKO$_{Ctsk}$ females of the control littermates. Immunohistochemical staining for Igfbp4 was performed, followed by counter staining with TRAP and fast green. (E) Tibiae were collected from 16-week-old Phlpp1 cKO$_{Ctsk}$ females of the control littermates following ovariectomy. Immunohistochemical staining for Igfbp4 was performed, followed by counter staining with TRAP and fast green.

levels within Phlpp1$^{-/-}$ osteoclast progenitors (Fig 4C). We next assessed IGFBP4 expression by osteoclasts within intact 12-week-old female mice. IHC was performed using an antibody directed towards IGFBP4 and sections were then TRAP and fast green stained. While modest immunostaining was observed within wild type females, robust staining was observed by TRAP positive bone lining cells within Phlpp1 cKO$_{Ctsk}$ female mice (Fig 4D). This robust expression of IGFBP4 by Phlpp1 cKO$_{Ctsk}$ osteoclasts was markedly diminished following ovariectomy (Fig 4E).

### Enhanced ex vivo osteoclastogenesis by Phlpp1 deficiency is limited by Igfbp4 inhibition

To determine if altered osteoclastogenesis could be restored by limiting IGFBP4 levels, we utilized the Igfbp4 protease PAPP-A [22]. Bone marrow progenitors were collected from 4-6-week-old Phlpp1 cKO$_{Ctsk}$ female mice or their control littermates. On days 0 and 3 of osteoclastogenesis assays, cells were treated with 20 ng/ml rPAPP-A or vehicle (Fig 5). Phlpp1 deficiency enhanced osteoclast numbers in vitro, but this was attenuated by addition of PAPP-A (Fig 5A). Western blotting confirmed that increased IGFBP4 levels within Phlpp1 deficient osteoclasts were suppressed by PAPP-A (Fig 5B).

### Discussion

In this study we evaluated the functions of Phlpp1 within Ctsk-expressing cells in a model of enhanced bone resorption. We found that despite elevated bone mass in intact females,

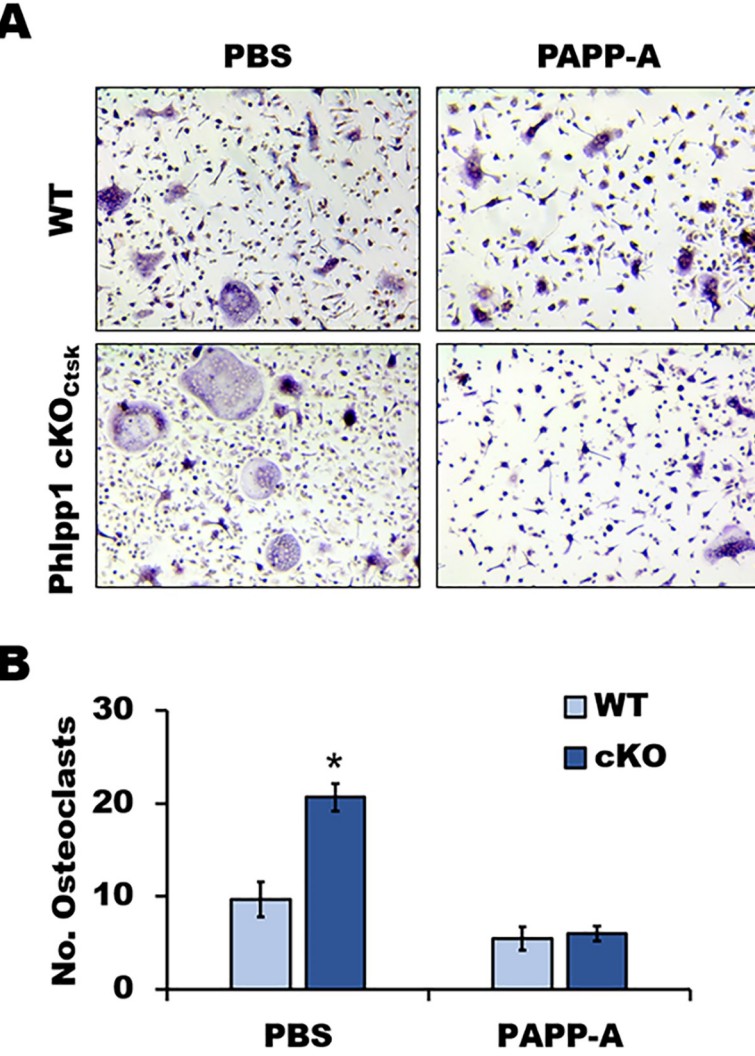

**Fig 5. Igfbp4 degradation attenuates enhanced osteoclast numbers induced by Phlpp1 deficiency in vitro.** (A, B) Osteoclasts were generated from 4-6-week-old female Phlpp1 cKO$_{Ctsk}$ mice or their control littermates. Cultures were treated with 20 ng/ml rPAPP-A or PBS on days 0 and 3. (A) TRAP staining was performed and (B) the number of osteoclasts in each condition was evaluated. *p<0.05.

Phlpp1 cKO$_{Ctsk}$ mice lose an equivalent amount of bone to their control littermates following ovariectomy. Moreover, we observed no differences in osteoclast number following ovariectomy. While this observation does not exclusively rule out the contributions of Ctsk-Cre expressing mesenchymal lineage cells, it supports that the effects are partially due to altered osteoclast activity. Ovariectomy also enhances osteoblast formation due to enhanced coupling of resorption to formation [23], but elevated osteoclast-mediated resorption outpaces this causing a net loss of bone mass. We did not see a change in osteoblast number or markers of bone formation following ovariectomy due to Phlpp1 ablation. Enhanced osteocyte apoptosis also occurs following ovariectomy, leading to enhanced bone resorption [24, 25], which could also be contributing to the phenotype. Future work is needed to further define the role of Phlpp1 in early osteoclast progenitor cells as well as osteoblast lineage cells.

In our study design, we estimated that we would need 7 mice in each group to detect changes in each group to provide 90% power to detect a 50% change in BV/TV ($\alpha = 0.05$), and

we were able to see significant declines in bone mass following ovariectomy. In contrast, the differences between ovariectomized Phlpp1 cKO$_{Ctsk}$ and Cre$^+$ littermates were slight. Given these differences, we would need significantly more mice (n = 26) within this group; thus, we conclude that bone mass following ovariectomy is not influenced by Phlpp1 expression within Ctsk-expressing cells.

In our work we focused on the effects of estrogen loss associated with ovariectomy. We show that Phlpp1 and Igfbp4 are inversely regulated by estradiol; thus, Phlpp1 may control Igfbp4 expression in an estrogen-dependent fashion. This is supported by the work of Farr et al. demonstrating that PHLPP1 levels were significantly diminished in specimens obtained from old women as compared to young women [26]. This decline in PHLPP1 expression observed with age was slightly restored by short-term estrogen therapy [26]. Prior work demonstrated that *Igfbp4* null female mice had low bone mass and increased osteoclast numbers compared to their control littermates [13, 27]. Moreover, *Igfbp4* null females were resistant to ovariectomy-induced bone loss [13, 27]. Given this, Igfbp4 was a prime candidate as it regulated bone mass in a sex-dependent fashion that was inverse to that of Phlpp1. Indeed, we find that limiting Igfbp4 levels mitigates enhanced osteoclast numbers of Phlpp1 cKO$_{Ctsk}$ cultures. Because of these data and prior publications, it is interesting to postulate a role for Igfbp4 as an estrogen-dependent coupling factor. In our analyses, we also identified other differentially expressed genes that contained ERE sites. These may also be regulated by Phlpp1 in an estrogen-dependent fashion and warrant future study.

Estradiol can elicit both genomic and non-genomic signaling, with the non-genomic pathway leading to activation of kinases including Akt and MEK/ERK [28]. We tested the ability of Phlpp1 to modulate activation of Akt and ERK1/2 in response to estradiol, but did not observe an effect of Phlpp1 deficiency. As this was the case, we focused on the genomic estradiol signaling and identified 67 potential targets that were differentially regulated by Phlpp1 deficiency osteoclasts that also contained ERE consensus sequences.

Ovariectomy reduces levels of estrogen, but it also imparts many other systemic changes. While our data demonstrate that Phlpp1 is regulated by estradiol, the effects observed following ovariectomy could be via alternate mechanisms. Future experiments will be aimed at testing the requirement of estradiol genomic responses to alter Igfbp4 expression and to determine if restoring estrogen status following ovariectomy reconstitutes the phenotype of Phlpp1 cKO$_{Ctsk}$ females.

## Author Contributions

**Conceptualization:** Elizabeth W. Bradley.

**Data curation:** Marcelline K. Hanson, Ismael Y. Karkache, David H. H. Molstad, Andrew A. Norton, Elizabeth W. Bradley.

**Formal analysis:** Marcelline K. Hanson, Ismael Y. Karkache, David H. H. Molstad, Andrew A. Norton, Elizabeth W. Bradley.

**Funding acquisition:** Elizabeth W. Bradley.

**Investigation:** Marcelline K. Hanson, Ismael Y. Karkache, David H. H. Molstad, Andrew A. Norton, Kim C. Mansky, Elizabeth W. Bradley.

**Supervision:** Kim C. Mansky, Elizabeth W. Bradley.

**Writing – original draft:** Marcelline K. Hanson, Elizabeth W. Bradley.

**Writing – review & editing:** Marcelline K. Hanson, Ismael Y. Karkache, David H. H. Molstad, Andrew A. Norton, Kim C. Mansky, Elizabeth W. Bradley.

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
