## [Decision Letter · Decision Letter 0]

11 Feb 2021

PONE-D-20-36058

PHLPP1 is induced by Estrogen in osteoclasts and its loss in Ctsk-Expressing Cells Does Not Protect Against Ovariectomy-Induced Bone Loss

PLOS ONE

Dear Dr. Bradley,

Thank you for submitting your manuscript to PLOS ONE. After careful consideration, we feel that it has merit but does not fully meet PLOS ONE’s publication criteria as it currently stands. Therefore, we invite you to submit a revised version of the manuscript that addresses the points raised during the review process.

We look forward to receiving your revised manuscript.

Kind regards,

Jung-Eun Kim

Academic Editor

PLOS ONE

Journal Requirements:

2.PLOS ONE now requires that authors provide the original uncropped and unadjusted images underlying all blot or gel results reported in a submission’s figures or Supporting Information files. This policy and the journal’s other requirements for blot/gel reporting and figure preparation are described in detail at https://journals.plos.org/plosone/s/figures#loc-blot-and-gel-reporting-requirements and https://journals.plos.org/plosone/s/figures#loc-preparing-figures-from-image-files. When you submit your revised manuscript, please ensure that your figures adhere fully to these guidelines and provide the original underlying images for all blot or gel data reported in your submission. See the following link for instructions on providing the original image data: https://journals.plos.org/plosone/s/figures#loc-original-images-for-blots-and-gels.

3. Please include your statement regarding the methods of anaesthesia and euthanasia in the manuscript Methods.

4.At this time, we request that you  please report additional details in your Methods section regarding animal care, as per our editorial guidelines:

(1) Please state the source and number of mice used in the study

(2) Please describe the post-operative care received by the animals, including the frequency of monitoring and the criteria used to assess animal health and well-being.

Thank you for your attention to these requests.

5. We note that you are reporting an analysis of a microarray, next-generation sequencing, or deep sequencing data set. PLOS requires that authors comply with field-specific standards for preparation, recording, and deposition of data in repositories appropriate to their field. Please upload these data to a stable, public repository (such as ArrayExpress, Gene Expression Omnibus (GEO), DNA Data Bank of Japan (DDBJ), NCBI GenBank, NCBI Sequence Read Archive, or EMBL Nucleotide Sequence Database (ENA)). In your revised cover letter, please provide the relevant accession numbers that may be used to access these data. For a full list of recommended repositories, see http://journals.plos.org/plosone/s/data-availability#loc-omics or http://journals.plos.org/plosone/s/data-availability#loc-sequencing.

7. Please note that PLOS does not permit references to “data not shown.” Authors should provide the relevant data within the manuscript, the Supporting Information files, or in a public repository. If the data are not a core part of the research study being presented, we ask that authors remove any references to these data.

8. Please ensure you have discussed any potential limitations of your study in the Discussion.

Reviewers' comments:

Reviewer's Responses to Questions

**Comments to the Author**

1. Is the manuscript technically sound, and do the data support the conclusions?

Reviewer #1: Yes

Reviewer #2: Partly

2. Has the statistical analysis been performed appropriately and rigorously? 

Reviewer #1: N/A

Reviewer #2: Yes

3. Have the authors made all data underlying the findings in their manuscript fully available?

Reviewer #1: Yes

Reviewer #2: No

4. Is the manuscript presented in an intelligible fashion and written in standard English?

Reviewer #1: Yes

Reviewer #2: Yes

5. Review Comments to the Author

Reviewer #1: In the manuscript “Phlpp1 is induced by Estrogen in osteoclasts and its loss in Ctsk-Expressing Cells Does Not Protect Against Ovariectomy-Induced Bone Loss” Hassan MK et al have described the influence of Phlpp1 in osteoclast activity in overiectomy mouse model in presence or absence of estrogen. The work might have a good contribution in osteoclast biology. However, the following issues need to be addressed for its improvement

1. Authors might explain about the “Ctsk-Cre expressing cells”, and full form of CTSK

2. Full form of IGFBP4

3. Author should explain the procedure of isolation of osteoclast progenitor cells and its purity.

4. In figure 5, authors may show representative photos of TRAP positive osteoclast cells.

5. In figure 1, it would be nice if authors can show few photos of scanned via micro-CT

6. In figure 4, authors may provide the densitometry data of western blotting.

Reviewer #2: In this manuscript, Hanson et al. document the effect of Ctsk-conditional Phlpp1 deletion on OVX-bone loss, and identify a potential mechanism by which estrogen regulates Phlpp1 and Igfbp4 expression. The underlying mechanism is very intriguing, however, there is a major issue with the overall premise. Specifically, since estrogen has effects on other bone cell lineages, including the osteoblast/osteocyte lineage, a loss of the increased bone phenotype with OVX does not specify a predominant role of Phlpp1 in resorption as compared to bone formation. In addition, there was no assessment of bone formation parameters in this model, beyond BV/TV, to support a lack of effect on coupling/bone formation. There are additional in vitro experiments/data that would strongly benefit this study.

Major comments:

1. As stated above, the rationale behind using OVX to test specificity for a resorptive vs formation effect of the Phlpp1 Ctsk-cre KO is flawed, as estrogen has effects on the osteoblast/osteocyte lineage, and OVX is known to impact osteoblast differentiation/bone formation.

2. The authors show there is no significant difference in bone phenotype between OVX WT and Ctsk Cre Phlpp1 KO animals. However, given that the sham phenotype is different, the authors should also assess percent change compared to sham. The fact that the animals have the same bone phenotype following OVX would suggest that the there was a more negative effect of OVX on bone mass in KO as compared to WT animals.

3. The authors showed that E2 increase Phlpp1 and decrease Igfbp4 protein levels and that Phlpp1 KO osteoclasts exhibited increased Igfbp4 expression. In order to fully understand this relationship and the impact on bone phenotype, the authors should do the same time course E2 treatment on WT and Phlpp1 KO osteoclasts.

4. In order to assess the impact of Ctsk Cre Phlpp1 KO or OVX on bone formation, the authors should assess dynamic bone formation rates and osteoblast number by histomorphometry. Current data only are reflective of the osteoclast data. This study would also benefit from serum markers of resorption and formation.

5. Given that osteoclasts could secrete IGFBP4, could this be a potential E2 induced coupling factor that increases bone formation in the sham mice?

6. PLOS authors have the option to publish the peer review history of their article (what does this mean?). If published, this will include your full peer review and any attached files.

Reviewer #1: No

Reviewer #2: No

---

## [Author Response · Author response to Decision Letter 0]

12 Apr 2021

Reviewer #1: In the manuscript “Phlpp1 is induced by Estrogen in osteoclasts and its loss in Ctsk-Expressing Cells Does Not Protect Against Ovariectomy-Induced Bone Loss” Hassan MK et al have described the influence of Phlpp1 in osteoclast activity in overiectomy mouse model in presence or absence of estrogen. The work might have a good contribution in osteoclast biology. However, the following issues need to be addressed for its improvement 

1. Authors might explain about the “Ctsk-Cre expressing cells”, and full form of CTSK

We delineated the Ctsk abbreviation within the manuscript; Ctsk-Cre expressing cells is used because the Ctsk-Cre transgene is expressed by different cell types.

2. Full form of IGFBP4

We defined Igfbp4 within the manuscript.

3. Author should explain the procedure of isolation of osteoclast progenitor cells and its purity.

We added this detail to the methods section of our manuscript.

4. In figure 5, authors may show representative photos of TRAP positive osteoclast cells.

We added these images to Figure 5.

5. In figure 1, it would be nice if authors can show few photos of scanned via micro-CT

Unfortunately, we do not have current access to some of the scans needed to generate reconstructions. 

Reviewer #2: In this manuscript, Hanson et al. document the effect of Ctsk-conditional Phlpp1 deletion on OVX-bone loss, and identify a potential mechanism by which estrogen regulates Phlpp1 and Igfbp4 expression. The underlying mechanism is very intriguing, however, there is a major issue with the overall premise. Specifically, since estrogen has effects on other bone cell lineages, including the osteoblast/osteocyte lineage, a loss of the increased bone phenotype with OVX does not specify a predominant role of Phlpp1 in resorption as compared to bone formation. In addition, there was no assessment of bone formation parameters in this model, beyond BV/TV, to support a lack of effect on coupling/bone formation. There are additional in vitro experiments/data that would strongly benefit this study.

Major comments:

1. As stated above, the rationale behind using OVX to test specificity for a resorptive vs formation effect of the Phlpp1 Ctsk-cre KO is flawed, as estrogen has effects on the osteoblast/osteocyte lineage, and OVX is known to impact osteoblast differentiation/bone formation.

We revised our manuscript to reflect the Reviewer’s comments and added discussion of this limitation to our revised manuscript. 

2. The authors show there is no significant difference in bone phenotype between OVX WT and Ctsk Cre Phlpp1 KO animals. However, given that the sham phenotype is different, the authors should also assess percent change compared to sham. The fact that the animals have the same bone phenotype following OVX would suggest that the there was a more negative effect of OVX on bone mass in KO as compared to WT animals.

We added these comparisons to Table 1 and discussed them within our revised manuscript. 

3. The authors showed that E2 increase Phlpp1 and decrease Igfbp4 protein levels and that Phlpp1 KO osteoclasts exhibited increased Igfbp4 expression. In order to fully understand this relationship and the impact on bone phenotype, the authors should do the same time course E2 treatment on WT and Phlpp1 KO osteoclasts.

We added these data to Figure 4 and discussed them within our revised manuscript.

4. In order to assess the impact of Ctsk Cre Phlpp1 KO or OVX on bone formation, the authors should assess dynamic bone formation rates and osteoblast number by histomorphometry. Current data only are reflective of the osteoclast data. This study would also benefit from serum markers of resorption and formation.

We added these data to our manuscript.

5. Given that osteoclasts could secrete IGFBP4, could this be a potential E2 induced coupling factor that increases bone formation in the sham mice?

We added this provocative idea to our discussion.

---

## [Decision Letter · Decision Letter 1]

3 May 2021

PHLPP1 is Induced by Estrogen in Osteoclasts and its Loss in Ctsk-Expressing Cells Does Not Protect Against Ovariectomy-Induced Bone Loss

PONE-D-20-36058R1

Dear Dr. Bradley,

We’re pleased to inform you that your manuscript has been judged scientifically suitable for publication and will be formally accepted for publication once it meets all outstanding technical requirements.

Kind regards,

Jung-Eun Kim

Academic Editor

PLOS ONE

Additional Editor Comments (optional):

Reviewers' comments:

Reviewer's Responses to Questions

**Comments to the Author**

1. If the authors have adequately addressed your comments raised in a previous round of review and you feel that this manuscript is now acceptable for publication, you may indicate that here to bypass the “Comments to the Author” section, enter your conflict of interest statement in the “Confidential to Editor” section, and submit your "Accept" recommendation.

Reviewer #1: All comments have been addressed

Reviewer #2: All comments have been addressed

2. Is the manuscript technically sound, and do the data support the conclusions?

Reviewer #1: Yes

Reviewer #2: Yes

3. Has the statistical analysis been performed appropriately and rigorously? 

Reviewer #1: Yes

Reviewer #2: Yes

4. Have the authors made all data underlying the findings in their manuscript fully available?

Reviewer #1: Yes

Reviewer #2: Yes

5. Is the manuscript presented in an intelligible fashion and written in standard English?

Reviewer #1: Yes

Reviewer #2: Yes

6. Review Comments to the Author

Reviewer #1: Authors have addressed queries raised by reviewer. This revision has improved the quality of the manuscript.

Reviewer #2: The authors addressed all of my comments. However, I don't see the edited figures in the resubmission.

7. PLOS authors have the option to publish the peer review history of their article (what does this mean?). If published, this will include your full peer review and any attached files.

Reviewer #1: No

Reviewer #2: No

---

## [Editor Report · Acceptance letter]

8 Jun 2021

PONE-D-20-36058R1 

Phlpp1 is Induced by Estrogen in Osteoclasts and its Loss in Ctsk-Expressing Cells Does Not Protect Against Ovariectomy-Induced Bone Loss 

Dear Dr. Bradley:

I'm pleased to inform you that your manuscript has been deemed suitable for publication in PLOS ONE. Congratulations! Your manuscript is now with our production department. 

Kind regards, 

on behalf of

Dr Jung-Eun Kim 

Academic Editor

PLOS ONE